# Glass Surface Nanostructuring by Soft Lithography and Chemical Etching

**DOI:** 10.3390/nano14211714

**Published:** 2024-10-27

**Authors:** Luciano Bravo, Martín Ampuero, Jonathan Correa-Puerta, Tomás P. Corrales, Sofía Flores, Benjamín Schleyer, Natalia Hassan, Patricio Häberle, Ricardo Henríquez, Valeria del Campo

**Affiliations:** 1Departamento de Física, Universidad Técnica Federico Santa María, Av. España 1680, Valparaíso 2390123, Chile; luciano.bravo@usm.cl (L.B.); martin.ampuero@usm.cl (M.A.); jonathan.correa@usm.cl (J.C.-P.); tomas.corrales@usm.cl (T.P.C.); sofia.floresb@sansano.usm.cl (S.F.); benjamin.schleyer@usm.cl (B.S.); patricio.haberle@usm.cl (P.H.); ricardo.henriquez@usm.cl (R.H.); 2Millenium Nucleus in NanoBioPhysics (NNBP), Av. España 1680, Valparaíso 2390123, Chile; nhassan@utem.cl; 3Instituto Universitario de Investigación y Desarrollo Tecnológico (IDT), Universidad Tecnológica Metropolitana, Santiago 8330383, Chile

**Keywords:** glass, nanotexturing, wet chemical etching, wetting

## Abstract

Due to its high durability and transparency, soda lime glass holds a huge potential for several applications such as photovoltaics, optical instrumentation and biomedical devices, among others. The different technologies request specific properties, which can be enhanced through the modification of the surface morphology with a nanopattern. Here, we report a simple method to nanostructure a glass surface with soft lithography and wet-chemical etching in potassium hydroxide (KOH) solutions. Glass samples etched with a polymeric mask showed a nanopattern with stripes of widths between 220 and 450 nm and modulated heights between 50 and 200 nm. For different solution concentrations or etching times, the obtained nanopatterns led to an increase or reduction of the water contact angle. The largest increment, ~20 degrees, was achieved by etching the glass for 180 min with 30% KOH concentration, while a super-hydrophilic glass (~9° contact angle) was achieved when etching for 90 min with the same concentration. Optical characterization showed a very low influence of the nanostructured pattern on glass transparency and an increment in UV transmittance for some cases.

## 1. Introduction

Self-cleaning surfaces have potential applications in biotechnology, instruments design and infrastructure maintenance [1,2]. Some of these uses, also require a high transparency, like in the case of photovoltaic solar modules, which are covered with glass to protect the solar cells [3]. To achieve self-cleaning hydrophobic glass, different coatings have been suggested; however, under outdoor conditions they have shown abrasion damage and degradation under UV exposure [4,5]. Conventional methods to achieve transparent self-cleaning surfaces use perfluoroalkyl substances [6]. However, there are growing concerns with respect to these chemicals, which exhibit signs of bioaccumulation that could lead to health issues [7]. Ways of achieving self-cleaning transparent surfaces using fluorine-free coatings are currently being developed using PDMS nanoparticles [8]. Hegner et al. reported fluorine-free silica nanoparticles coated with PDMS to achieve super-repellency, which is a fundamental property of self-cleaning surfaces [9]. Nevertheless, all these methods require coating the glass surface with other chemicals to modify the wetting properties. Thus, instead of coating, the direct modification of the glass surface could be a more robust solution to modify the wetting properties of a surface.

A standard procedure to nanostructure glass surfaces is through the deposition of a thin metal film (between 4 and 15 nm) that is annealed to form nanoparticles on the surface to conform the mask for a dry etching process [10,11,12,13]. Glasses with this mask are submitted to reactive ion etching (RIE) after which nanopillars or nanocones form on the glass surface with different geometries that depend on etching time and power. With these 2D disordered patterns, the glass surface becomes more hydrophilic, reducing the water contact angle to values between 2.3 and 46 degrees.

Baquedano et al. [14] transferred the 1D (nanowires) pattern of a Digital Versatile Disc (DVD) onto a glass surface through soft lithography followed by RIE with CHF_3_. Soft lithography enables the replication of a desired pattern using an elastomeric stamp onto which the pattern is transferred. Then, the stamp is pressed against a polymer that covers the sample until it dries. In this way, the polymer becomes a mask that allows etching the specific pattern onto the sample [15]. The glass surface was nanostructured with a linear pattern of 469 nm width and 23 nm height, which increased the water contact angle from 63.4° to 91.3°. In contrast, their 2D disordered nanostructure decreased the contact angle to 30°. Depending on the application and maintenance protocols, superhydrophilicity is also useful for achieving self-cleaning properties [16].

Regarding wet etching for surface nanostructuring, potassium hydroxide (KOH) is widely used for anisotropic texturing of silicon solar cells [17]. Canavese et al. [18] studied the effect of different KOH concentrations (between 20% and 40%) in Si and SiO_2_ etching rate. Their results show that when the KOH concentration increased, the etching rate of silicon substrates decreased, but the ratio between SiO_2_ and Si etching rates increased.

In this work, we used a combination of soft lithography and KOH wet etching to structure the surface of soda lime glass with a 1D pattern (nanostripes). In this way, we were able to transfer the pattern of a DVD (line width 400 nm) by using polymethyl methacrylate (PMMA) as a protective mask and KOH as an etchant. KOH has already been used, for a similar purpose, in the manufacture of solar cells for antireflecting texturing of silicon wafers [17]. Here, we report the effect of solution concentration and immersion time on the wetting and optical properties of a glass substrate.

## 2. Methods

In this study, we used soda lime glass due to its high transparency and durability and extended use in photovoltaic modules among other applications. The patterning process was performed through soft lithography followed by wet etching in a KOH solution. We washed the glass samples—microscope slides cut in chips of 1 × 1 cm^2^ (1 mm thickness)—with water and detergent and then sonicated them in three different solvents (4 min each): deionized water, isopropanol, and ethanol. The mold for soft lithography was a DVD which contained the stripe pattern we wanted to transfer to the glass. Through atomic force microscopy (Figure 1a,b), we determined the stripes’ height to be 88 ± 2 nm and a width of 378 ± 7 nm (at half height), with a periodicity of 765 ± 7 nm.

To transfer the pattern, a 3 mm thick layer of polydimethylsiloxane (PDMS) was deposited on the DVD to obtain an elastomeric stamp of the structure. Square stamps of 2 cm^2^ were cut from the DVD to ensure a complete transfer of the pattern to the final sample. The glass chip was covered with a fresh layer of polymethyl methacrylate (PMMA). For this purpose, we dissolved 4 g of PMMA (molecular weight 100.115 g/mol) in 96 mL of toluene with magnetic stirrer for 72 h. A 1 µL droplet of this solution is deposited on the glass and then patterned by pressing the stamp on top. To ensure a homogeneous pressure, we designed a press with an area of several square centimeters with a homogenous weight of 1.5 kg. After 5 min, the PMMA was dry, and the optical diffraction of the DVD was observed in the PMMA mask. AFM measurements of the PMMA mask on the glass chip (Figure 1c,d) show a striped pattern with a geometry like that of the DVD, except for the stripes’ height, 34 ± 2 nm.

The glass chips with the PMMA mask were introduced into KOH solutions with concentrations between 20% and 40% at 70 °C. Inside the solution, the samples were placed on a Teflon stage similar to that designed by Canavese et al. [18]. The etching process in KOH varied between 90 and 240 min [18]. After etching, the PMMA mask was removed with acetone, and the glass chip was cleaned again.

Morphological characterization was performed through field emission scanning electron microscopy (FESEM, Zeiss GeminiSEM 360, Oberkochen, Germany) and atomic force microscopy (AFM, Flex AFM, Nanosurf, Langen, Germany). Water contact angle measurements were performed with an Attension^®^ Theta Optical Tensiometers (Biolin Scientific AB, Gothenburg, Sweden) using a 7 µL drop. The contact angle is automatically given by the OneAttension^®^ 1.8 software, which allows choosing the fitting model. In this case, we chose the Young–Laplace model, and the reported results corresponded to the average between left and right angles and the measurement of 3 different areas. The samples’ optical transmittance was determined by UV-visible spectrophotometry (Olis Cary 14) between 298 and 800 nm. This equipment uses a Cermax Xenon arc lamp as light source and photomultiplier (PMT—Electron Tubes 9828WB, Athens, GA, USA) as detector.

## 3. Results and Discussion

### 3.1. Nanostructured Glass

After the patterning process, the striped structure of a DVD was successfully transferred to the glass surface, as shown in Figure 2a. Using the program ImageJ, we analyzed the different geometrical parameters of the resulting structure. The width of the stripes varied between 250 and 450 nm, values that are consistent with the separation between the stripes measured in the PMMA mask (Figure 1d). The black square in Figure 2a delineates an area of the same size as the one shown in the atomic force microscopy (AFM) image of Figure 2b. Red lines were drawn to highlight some of the transferred stripes from the DVD structure. White lines (x_1_ to x_6_) indicate some locations on which line profiles of the stripes were measured (other profiles were taken from other areas of the sample); the resulting average is shown in Figure 2c. To analyze the stripes topography in detail, we fitted two Gaussian peaks, plotted as the blue dashed line. Through the full width at half maximum, we obtained a stripes’ width of 272 ± 36 nm; however, measuring different sets of profiles indicate the width of the stripes can vary between 220 and 450 nm. These values are in good agreement with the SEM measurements. The Gaussian fit to the stripe profile was consistent with an average height of 65 ± 10 nm, while measuring in different sections of the sample, indicated a height variation between 50 and 200 nm. We compared the line profile of a valley (white line y1) with that on top of the stripe (white line y2). Figure 1d shows fluctuations in the height along the stripes, which could be described as the formation of mounds. The height of these mounds varied between 20 and 180 nm, resulting in a non-uniform height of the glass stripes. The valley’s profile (red line in Figure 2d) also showed mounds but with lower height than the stripes’ summit.

Our results indicate that the 1D patterns transferred to the glass surface after wet etching with KOH over a PMMA mask led to a striped pattern, confirmed by SEM and AFM (Figure 2a,b). The stripes’ roughness and width variation are probably due to a limited mask selectivity, as the PMMA is partly attacked by KOH, and to the width and height modulations of the PMMA mask. In the next sections, we report the effect of KOH concentration and etching time on the water wetting of the surface and the modification of its optical properties.

### 3.2. Wetting Properties

#### 3.2.1. Experimental Results

Figure 3 shows the measured contact angle of the bare glass and nanostructured samples. For the bare soda lime glass, we measured a contact angle of 43 ± 2 degrees, which is plotted with a hatched area (oblique lines pattern) for comparison purposes. The other hatched area (squares pattern) corresponds to the contact angle of a glass chip submitted to KOH etching for 180 min without mask (38 ± 2 degrees).

For the nanostructured samples, we analyzed the contact angle as a function of KOH concentration for a fixed etching time of 180 min (results are shown in Figure 3a). When the etching process was performed with a KOH concentration of 20%, the glass contact angle increased to 49 ± 2°, while a larger increment was observed for a concentration of 30%, reaching 61 degrees. Conversely, the surface contact angle was reduced to 23 ± 2 and 29 ± 2 degrees for concentrations of 25% and 40%, respectively. As expected, nanostructuring the surface modified the glass-wetting properties, but there seemed to be no correlation between the KOH concentration and wetting properties as determined by measuring the water contact angle.

To analyze the effect of etching time (Figure 3b), we chose a KOH concentration of 30%, since that sample showed the largest difference of the contact angle with respect to the bare glass. A first batch of samples was subjected to etching times between 30 and 210 min, while a second batch was treated for etching times above 120 min. Glass immersion in the KOH solution for 30 min resulted in a reduction of the surface contact angle to 22°, which slightly decayed to 20.5° with 60 min and reached a minimum of 9.3 ± 2° with 90 min of etching. Then, the contact angle started increasing until, at 180 min, maximum values were achieved (61.4 ± 2.3° and 60.1 ± 1° for the first and second batch, respectively). After 120 min of etching, the contact angle was again reduced to values between 20 and 24 degrees. As observed by changing KOH concentration, there was no correlation between etching time and contact angle.

Baquedano et al. [19] transferred the pattern of a DVD onto a silicon wafer through soft lithography and RIE. They used a PDMS stamp and PMMA layer to transfer the pattern onto an SiO_x_ layer, which is used as a hard mask for RIE etching of the silicon surface with SF6. They etched all samples for 1 min and analyzed the surface topography and contact angle as a function of the RIE power (between 10 and 90 W). Their results showed some correlation between the stripes’ aspect ratio (depth/linewidth) and the RIE power; however, there was no correlation between the water contact angle and the RIE power. The use of RIE (combination of ionic and chemical attack) cannot be directly compared to the wet etching with KOH process used in our samples (purely chemical), as ways to modify the surface wettability properties. However, it is interesting that in both cases there was a lack of correlation between the etching parameter (RIE power or wet-etching time) and the water contact angle. The increment in the etching rate could either enlarge or reduce the contact angle.

#### 3.2.2. Theoretical Aspects

The increment or reduction of the contact angle depends on the wetting regime, which is correlated with the surface’s roughness. If the droplet penetrates the surface grooves, the phenomenon is described by the Wenzel regime [20] (Equation (1)).
(1)$$\cos\theta_{c}=r\cos\theta_{y}$$
where $\theta_{y}$ is the contact angle of the smooth surface, $\theta_{c}$ the measured contact angle (of the rough sample) and r is the roughness factor, defined as the ratio between the real surface area and the projected (flat) area. Since r must be larger than 1 (r≥1), if $\theta_{y}<90{}^{\circ}$ then $\theta_{c}$ must be lower than $\theta_{y}$. Although this model was initially derived for a surface with isotropic roughness, it also applies to any other patterned surface as long as the droplet wets all the surface below it.

If the surface grooves are small enough and the water droplet cannot penetrate within them, we are in the presence of the Cassie–Baxter regime with a roughness term, modeled by Equation (2) [21,22,23,24,25].
(2)$$\cos\theta_{c}=f\left(r_{f}\cos\theta_{y}+1\right)-1$$
where f is the ratio between the solid area in contact with the water and the flat surface, while 1−f corresponds to the ratio between the air–liquid area below the droplet and the flat surface. The term $r_{f}$, proposed by Marmur et al., is the roughness ratio of the surface in contact with the liquid [25].

In this work, most samples showed a reduction in the contact angle, so they were in the Wenzel regime. Although the bare glass surface was not completely flat ($r_{rms}=0.6\;\mathrm{nm}$), we considered the measured angle for this surface as $\theta_{y}$ (43°) to analyze the modification in surface roughness due to KOH etching. For samples etched with 30% KOH for 30, 60, 120, 150, 210 and 240 min, the calculated roughness factor was r=1.3. The glass etched with 40% KOH (for 180 min) showed the lowest reduction in the contact angle, Δθ = 14°, which implied r=1.2. The largest roughness factor, r=1.4, was achieved by the glass etched for 90 min (30% KOH), which also showed the lowest contact angle (~9°).

To evaluate the applicability of the Wenzel model in this system, we analyzed glass samples etched without mask through the water contact angle and AFM measurements. The contact angle after etching was $\theta_{c}=38{}^{\circ}$, which, according to the Wenzel model, implies a roughness factor r=1.1. This means that after the texturing process without mask, the surface area in contact with water increased by ~10%. Figure 4 shows AFM images of the glass surface before (Figure 4a) and after KOH etching (Figure 4b). According to the AFM analysis, the surface area increased from 100 µm^2^ to 100.3 µm^2^, which is only 0.3%. This value is too low in comparison to the results from the Wenzel model. Thus, in this case, there was no direct relationship between the increment in surface area and the change in the contact angle. The Wenzel model, though initially derived for a surface with isotropic roughness, also applies to any other patterned surface but only if the droplet provides homogeneous wetting of the surface below. Possibly for the case of a porous surface (Figure 4b) or a complex nanometric pattern (Figure 2b), additional parameters have to be considered [26].

An even more unexpected scenario was the increment of the contact angle observed after etching for 180 min with KOH concentrations of 20 and 30% (Figure 3a). If we were to use the Wenzel model in this case, we would find a roughness factor below one, which is not possible. Through AFM measurements (Figure 2b), we found that the ratio between rough and flat surface area was 1.1 for the sample etched for 180 min with 20% KOH. This sample had a contact angle of 49.2°, but if we use r=1.1 in the Wenzel model, the contact angle gives ~36°—a lower value than the $\theta_{y}$, as expected for this model. Conversely, the Cassie–Baxter regime explains increments of contact angles but only for hydrophobic surfaces (with $\theta_{y}>90{}^{\circ}$) or for hydrophilic ones in which the structure has undercuts [27].

Given our findings, we believe neither the Cassie–Baxter nor Wenzel models fully describe all our contact angle measurements, which show water contact angle increased or decreased depending on etching protocol. Given that all surfaces in this study were hydrophilic, our results could indicate the presence of hierarchical structuring of the surface, including two length scales: micron-patterned stripes and nanometric pores on glass. Contact angles over hierarchical structures were theoretically described by Herminghaus [28], who showed they could increase even when the textured surface was originally hydrophilic. 

### 3.3. Optical Properties

We studied the effect of surface nanostructuring on the glass optical transmittance between 298 and 800 nm. Figure 5a shows the transmittance measurements for the bare glass (black line) and for samples treated for 180 min with KOH concentrations varying between 20% and 40%. In all cases, transmittance in the visible range was lower than that of the bare glass. Samples prepared with KOH at 20, 25 and 40% showed very similar curves between ~350 and ~570 nm. Above ~600 nm, the sample prepared with the lowest concentration showed a positive slope (blue line), which was also the case for 25% of KOH (light blue line) but with a smaller increment. The sample with the lowest transmittance was the one prepared with 30% KOH (red line), which was also the one with the largest water contact angle. Transmittance of the nanostructured glass with KOH concentration of 20% showed a blueshift of 10 nm, enabling larger light transmission in the UV region. Although this shift was not observed for KOH concentrations above 20%, in all cases transmittance near 300 nm was larger than for the bare glass.

Figure 5b shows the transmittance measurements of the bare glass (black line) and patterned glass etched with a KOH concentration of 30% for different times. The sample processed for 90 min (blue line) also showed the blueshift observed for the glass patterned with the minimum concentration (20% KOH for 180 min). The transmittance curve of the glass processed for only two hours (light blue line) showed a broad valley with its minimum at 587 nm. Among the nanostructured glasses, two samples call our attention. The sample with the lowest contact angles (30% KOH for 90 min) was also the one with the highest transmittance curve, while the one with the largest contact angle (30% KOH for 180 min) showed the lowest transmittance.

The total transmittance of each sample was computed as the area below the curve divided by the area of a theoretical transmittance (100%), calculated between 298 and 800 nm (∆λ=502 nm), as shown in Equation (3):(3)$$T_{total}=\frac{\int_{298}^{800}T\left(\lambda \right)d\lambda }{100\;∆\lambda }$$
where $T\left(\lambda \right)$ is the sample’s measured transmittance. Figure 5c shows the total transmittance of each sample vs. the KOH concentration of the solution. The total transmittance of the bare glass was 93.2% and is plotted as a horizontal line for comparison purposes. Independent of the KOH concentration used in the etching procedure, glass transmittance was always close to 90%, with a minimum value of 89% observed for 30% KOH concentration. Except for this sample, glass transmittance had values between 90.4% and 91.8%. Figure 5d shows total transmittance as a function of etching time. In this case, the highest transmittance was 91.1% for an etching time of 90 min, which abruptly decayed to 87.2% with 120 min of processing and increased again to 90.9% for 150 min exposure. The lowest transparency observed for the 120 min sample was related to the valley shown by the optical measurement (Figure 5b). Samples etched above 180 min had transmittance values of ~89%.

In all our samples, the glass texturing with a DVD pattern decreased the optical transmittance. The opposite effect, an increment in glass transmittance, was reported by Baquedano et al. [14] when they transferred the DVD pattern through RIE. This difference could be explained by the stripes’ height, which in their case was ~20 nm and in our samples was ~100 nm, or by the large roughness of our stripes. Indeed, the 2D nanostructures developed in their work (with height of ~200 nm) decreased the glass transparency.

## 4. Conclusions

In this work, we achieved the transfer of a nanopattern to the surface of soda lime glass through the integration of soft lithography with KOH wet etching. The pattern consisted of stripes with variable heights between 50 and 200 nm and widths between 220 and 450 nm. This surface modification did not compromise the glass transparency since optical transmittance under different process conditions remained mostly above 90%.

We analyzed the effect of the glass-texturing process on its water contact angle as a function of KOH concentration and etching time. We observed an increment in the contact angle for two concentrations, 20% and 30%, and a decay for 25% and 40% KOH. Regarding etching time, the contact angle decreased in most cases, except when the glass was in the KOH solution for 180 min. In this case an increment of ~20° with respect to the untreated surface was achieved, which is similar to that obtained by RIE [14]. Conversely, the same KOH concentration (30%) applied for 90 min led to a superhydrophilic surface with a contact angle of ~9°. Hence, the change in the contact angle showed no correlation with KOH concentration or processing time.

For a hydrophilic material like glass, the change in water contact angle due to surface texturing is usually explained by the Wenzel model. In this regime, the increment in surface roughness (due to texturing) leads to a reduction of the contact angle, which was observed in most of our results. However, the calculated surface increase using the Wenzel equation did not necessarily agree with AFM measurements. Further, the observed increments in contact angle are unexpected in the Wenzel model, and the Cassie–Baxter regime only applies to hydrophobic surfaces or when the nanostructure has undercuts. Further studies combining experiments and theory are necessary for a deeper insight on the effect of surface nanostructuring on glasses’ wettability properties.

For applications, this paper presents a simple way to pattern a glass surface and modify its wetting properties with very low effect on transparency. This etching process could be easily used on photovoltaic glasses to reduce dust accumulation, since KOH is widely employed to texture silicon wafers. Further, self-cleaning glasses can also be used in everyday life objects, such as car windshields. Glass patterning and etching is also applied for MEMS (micro-electromechanical system) devices due to its electrical isolation properties and for biochips where specific wetting properties are desirable. Our results give further knowledge of how to achieve nanostructured glass surface, which could be used for the fabrication of suitable glass for different technologies.

## Figures and Tables

**Figure 1 nanomaterials-14-01714-f001:**
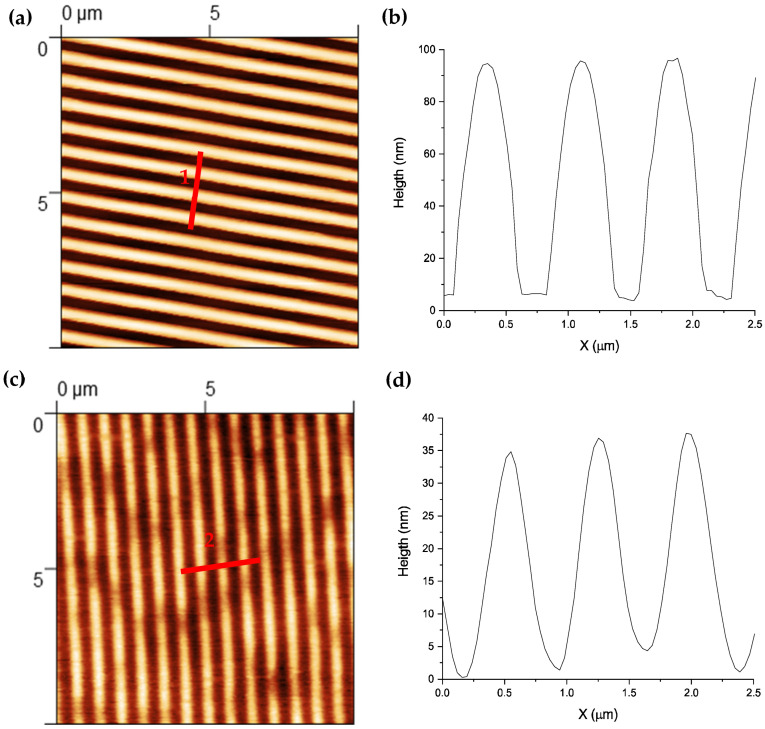
(**a**) AFM image of a DVD (10 × 10 µm); (**b**) height profile of the transversal cut shown by line 1; (**c**) AFM image of the PMMA mask on the glass chip after transferring the pattern with the PDMS stamp (before KOH etching); (**d**) profile of the transversal cut shown by line 2.

**Figure 2 nanomaterials-14-01714-f002:**
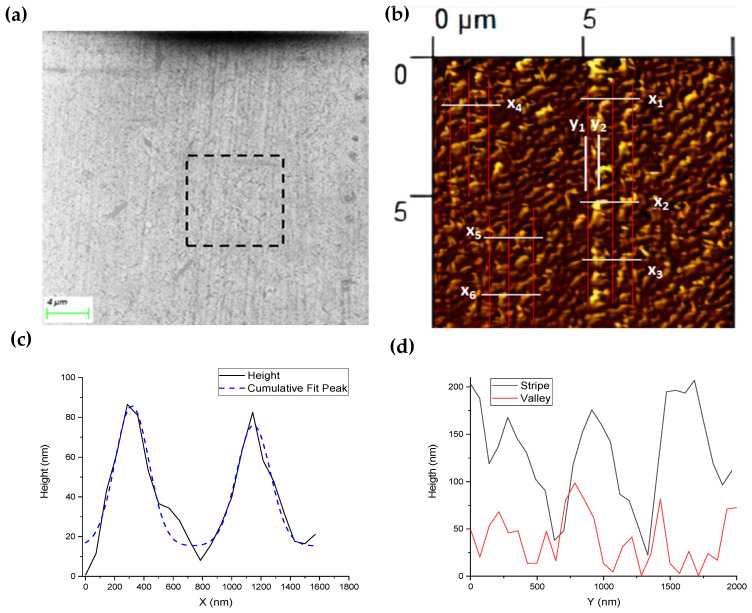
(**a**) SEM image of nanostructured glass (20% KOH during 180 min) with 2.5 kx magnification; the black square demarks an area of the size of the AFM measurement; (**b**) AFM topographic image of the nanostructured glass with red lines that indicate the stripes orientation; (**c**) profile obtained through the average of the transversal cuts (perpendicular to the stripes) shown by the black lines (x_1_ to x_6_) in the AFM image; and (**d**) profile of transversal cuts along a valley between stripes (y_1_) and the stripes summit (y_2_), indicated by the white vertical lines in the AFM image.

**Figure 3 nanomaterials-14-01714-f003:**
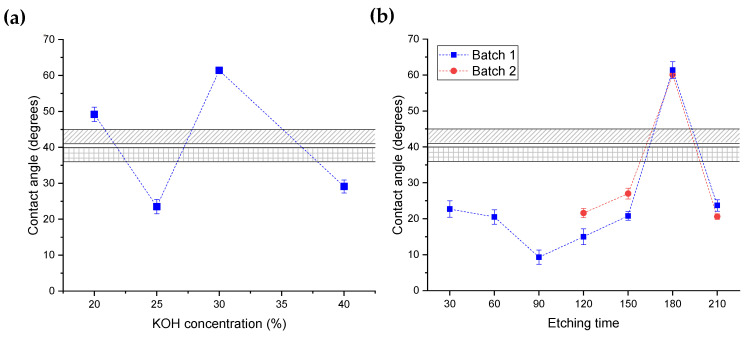
Contact angle of glass surfaces: bare glass (hatched area with lines), etched glass with no mask (hatched area with a square pattern), nanostructured glass surfaces (colored symbols linked by a dashed line). (**a**) Contact angles as a function of KOH concentration (etching time 180 min), and (**b**) contact angles as a function of etching time (KOH concentration of 30%) for two batches of samples.

**Figure 4 nanomaterials-14-01714-f004:**
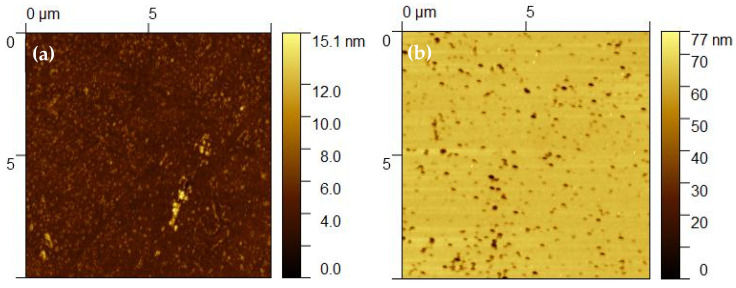
Atomic force microscopy (AFM) images of the glass surface (**a**) before etching and (**b**) after KOH etching for 30 min (30% concentration).

**Figure 5 nanomaterials-14-01714-f005:**
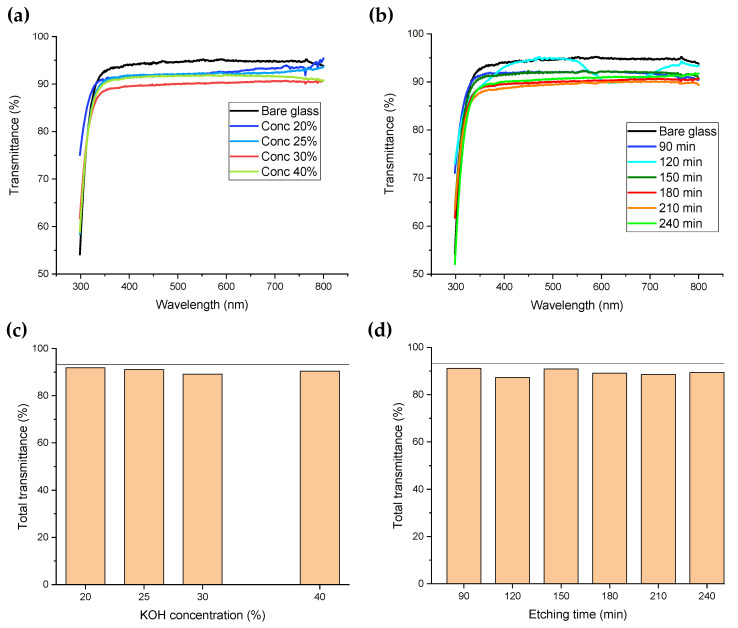
(**a**) Transmittance as a function of KOH concentration (180 min). Measurements performed between 298 and 800 nm for bare glass and structured surfaces with varying KOH concentrations: 20%, 25%, 30%, and 40%; (**b**) transmittance as a function of exposure time (KOH 30%). Measurements performed for 30, 60, 90, 120, 180 and 240 min of etching time; (**c**) total transmittance as a function of KOH concentration (180 min) and (**d**) total transmittance as a function of wet etching time (30% KOH). The black line corresponds to the total transmittance of the bare glass sample, 93.2%.

## Data Availability

Data is contained within the article.

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
