# Peer review of "Glass Surface Nanostructuring by Soft Lithography and Chemical Etching"

_nanomaterials, 2024, doi:10.3390/nano14211714_

Round 1
Reviewer 1 Report
Comments and Suggestions for Authors
Comments on nanomaterials-3156794-v1
„Glass surface nanostructuring by soft lithography and chemical etching“
The manuscript addresses nano-structuring of a glass surface for different purposes (improved and decreased wetting), where the surface is patterned by methods that are simple to perform (to have a good up-scaling potential), namely soft lithography and wet etching (KOH).
Parts of the manuscript are sound and show that the authors try to make use of ‘classical’ knowledge (Wenzel and Cassie-Baxter theory) in surface wetting to introduce their concept and to interpret their results.
However, the realization of their experiments and the structural results obtained lack in explanation and soundness, resulting in the urgent need for revision, mainly in the methods part and in the results described on page 3 to 5. In particular, it is far from apparent how the structures shown in Fig. 3e might develop.
Beyond that, the theory addressed is less suited to explain the results.
At the end of the comment a proposal is suggested for improvement of the manuscript.
These are the detailed questions to be answered and improvements of the text to be implemented:
Methods section
General: Choose one tense (present, past) and stay there.
Par 1
- Are the samples used microscope slides? (A comment on page 3 referring to Fig. 2 a/b suggests this) – add correct denomination of samples
Par 2
- Layer thickness of PDMS is missing! Is a backplane used? Is the PDMS cut to 1cm x 1 cm (similar to sample size), or is it larger/smaller? Further missing: supplier of PDMS, mixing ratio, curing conditions.
- PMMA layer preparation missing. Thickness? Application (droplet? spinning?, …)? Molecular weight of PMMA? Solvent? Concentration of solution?
- Soft-lithography details missing. Out-squeezing of excess solution? Pressing (by hand, by weight, with a counter-glass, …)? ‘Drying’ at room temperature (by in-diffusion into the PDMS)? Which indication exists for the drying to be completed?
- Do the authors assume, that in elevated parts of the PDMS stamp there is contact to the glass substrate (dewetting between elevated lines, no residual layer of PMMA remaining?) If yes, how is that proven?
- The last sentence of par 2 is completely unclear: Why should the height of the structures be determined by the thickness of the PMMA layer? I expect, the thickness is unclear (droplet), and then the height of the structures obtained depends mainly on the concentration of the PMMA in the solution (and to a lesser extent on the cavity height that might become decreased by pressing of the PDMS stamp)
- In case that sample and stamp are of differing size: is there a ‘reservoir’ of solution along the edge of the smaller part, where further PMMA solution diffuses into the linear cavities (when the solvent evaporates through the stamp)
Par 3
- Temperature of KOH solution missing
- Any expectation on etching rate of glass in KOH is missing (own tests?, literature?), otherwise the set of times (90-240 min) remains strange
- similarly, an expectation on isotropy of etching glass in KOH is missing. It should be possible for the reader to understand the parameter choice of the authors
- furthermore, what is the mask selectivity? This is an important parameter in order to understand, whether the PMMA is partly etched in the KOH (thus to address/understand the strange result obtained in Fig. 2…)
Par 4
- which method is used (may be automatically) for evaluating the contact angle from a droplet picture? (height-width, …, Laplace-fit) The different methods existing are best suited for different situations, and often provide different results
Results and discussion / nanostructured glass
Fig 2
- what is the reason to show two different magnifications? What information is provided? (not mentioned in text) In my view, the larger magnification is sufficient, and I would also skip the SEM of the un-patterned sample (a/b), as the authors cannot comment on the ‘clouds’ visible (proposal: show only d to g)
- in d) I would prefer to have the quare rotated, so that it corresponds with e). AND: scale bars should be readable
- in the text the authors mention that according to ImageJ the thickness (better: width?) of the stripes varies between 250 and 450 nm. This is a huge variation, and the image analysed (in case it is the one shown) is not really appropriate for this purpose. Why did the authors not take an image more suitable (meaning with more distinct structures)?
- What (??) is visible in e? without the labelling it would have been very difficult to identify the ‘lines’… There seem to be mounds of differing height (the authors say 50-200 nm, this is a 400% variation!), with ‘some’ direction of alignment. How can this correlate with the mask shown in Fig 1c/d? Potentially, the authors should add some more scans in Fig 1d, also across ‘darker’ parts of the lines, in order to have some indication what happened during the etching (erosion of mask during etching?). The result in Fig 2 e/f/g ultimately requires an explanation!
- Why the authors did not provide an AFM across areas similar to Fig 1 (10 x 10 µm) in order to improve the comparability between mask and etching result? (could also be an additional image) This is strongly suggested!
- The last sentence starting on page 3 is not understandable: why should the physical dimensions of the stripes depend on the glass etching rate, and why should this glass etching rate vary along the surface??? What do the authors mean really? Please modify!
- Last sentence of this paragraph: why is the variation of height along the lines (20-180 nm) different from the one across the lines (50-200 nm)? Both should be identical
- I propose to skip the phrase ‘interesting height variation’, and rather address reasons for it
- Caption: etching conditions (KOH concentration, etching time) are missing – this is important to ‘locate’ the results of Fig. 2 somewhere in the parameter space of Fig. 3! Is it 30%/180 min?
Results and discussion / wetting properties
Par 1
- How many measurements were taken for the contact angle? The static contact angle may vary, depending on a lot of issues, in particular at a surface as the one shown in Fig 2e! On such an inhomogeneous surface, different locations should be measured, and the respective measurement should go across the lines and along the lines, as both may differ (along the lines ‘wicking’ may occur). When the sample is too small for measuring different droplets aside (a dry surface is required for a new measurement) then several samples are required. Multiple measurement points across the surface are also advised for the un-patterned sample, as the ‘clouds’ in Fig 2a/b may indicate chemical non-uniformity. Give correct origin/formation of measured values shown! (The authors claim ‘no correlation’ of wetting properties and etching parameters, but go on and analyse the results in detail on page 6 – somewhat inconsistent)
- Before any interpretation, the authors should undoubtedly conform the measurements, in particular the point at 30%/180 min – the second batch in Fig 3b is a good attempt, but seems not to be sufficient (more values around ‘peak’, second batch in a, …)
- Is it proven, that the small decrease of contact angle after KOH etching without mask does NOT result from a different surface chemistry (compared to the virgin substrate)? Later on, the authors ascribe this to an increase in roughness, r = 1.1 – such a roughness (or roughness difference to the bare glass) should be measurable with the AFM! Confirm roughness by measurement!
Results and discussion / wetting properties discussion, page 6
- The ref Baquedano is number 14 (not 10)
- The investigation of Baquedano is hardly comparable with the results obtained by the authors (they used RIE, a plasma process, where ion impingement and thus directionality may contribute to the etching result – however, Baquedano does neither provide clear data/properties of his etching process or etching system). Anyway, some combination of directional and isotropic etching can be expected; by all means, his lines are more clearly defined after etching (see Fig 2 there)
- In this publication Baquedado did NOT vary the etching conditions!
- His result is that the contact angle can be reduced or increased by patterning the surface. However, his clear correlation is: increase with ordered (linear) structures, decrease with arbitrary structures. In consequence the first 9 lines of this paragraph have to be re-written/corrected
- Section on Wenzel and Cassie-Baxter in general / lines 11-29, page 6: Largely ok; however, the authors did not mention that Wenzel, though initially derived for a surface with isotropic roughness, also applies to any other patterned surface, as long as the droplet provides homogeneous (= complete) wetting of the surface below the droplet. In contrast, Cassie-Baxter exclusively refers to a periodic surface pattern, and features inhomogeneous wetting of the surface below the droplet (wetted and non-wetted parts)
- AND: I prefer the CB equ with an additional roughness factor (for the wetted part only) multiplied to the cos Theta-y-term – this refers to situations when the wetted part of the structures is NOT flat, which is the case in the original paper of CB (they addressed fibers with a circular cross section, a porous surface, and in their relationship the sum (f-1 + f-2) is larger than 1). Later, the CB-equation was re-written (by introducing the relative PROJECTED wetted area), see e.g. your ref 21, Marmur. This re-formulation simplifies the comparison/transition to Wenzel, when f = 1. When the structures have flat tops, the equation may be used without this roughness term (without clearly indicating this issue), as the roughness then is 1. However, the structures shown by the authors are far from providing a flat surface part on top, so skipping this factor is not justified. Please correct accordingly!
- Section on evaluation of experimental results, last two paragraphs on page 6: The authors behave, as if it were possible to use both descriptions, Wenzel (W) and Cassie-Baxter (CB) ‘in parallel’ and to apply W when the contact angle decreases but to apply CB when the contact angle increases. This is not fully correct. Which of both situations (W or CB) develops depends on the application method of the droplet, the structure type (holes or pillars) and in particular on the type of equilibrium (local or total) achieved. This issue is e.g. addressed in detail in Ref 19 and 20, e.g. Fig 5 in ref 20.
Generally, for lower contact angles W provides the lowest energy situation; only with very high contact angles theta-y (well above 90°, see Fig 5 in ref 20) CB provides the lowest energy.
In particular with values theta-y below 90° (as it is the case with the results presented here) CB can only lead to a local minimum, with an energy higher than the respective W state. AND, moreover, attaining such a local CB minimum at theta-y < 90° requires surface structures with an undercut, where the sidewall angle (when measured between bottom flat and sidewall) is low enough to pin the liquid along the edge of the upper wetted part of the structures. The limiting value is about sidewall angle ≤ theta-y. With a sidewall angle larger than 90° (as it seems to be the case, from Fig 2), even a meta-stable CB situation cannot be obtained, with the exception of hierarchical structures.
This is the issue to be solved/discussed with the actual results.
- Evaluation of roughness from W: should be done for all samples. As the authors have an AFM at hands it is a very simple task to assess the surface scan in terms of roughness, and to compare it with the ‘theoretical result’.
- Evaluation of wetted part f from CB: Is very critical, on three grounds. (1) With theta-y < 90° only a local minimum could be obtained, and this has a limited stability (meta-stable). It is hard to know, under which experimental situation it is reproducibly obtained. (2) An undercut structure (sidewall angle < about 40° here) is required to obtain a CB situation – undercuts/overhangs are not measurable by AFM. Indication for potential undercuts could only be given by SEM cross sections, which are hard to get with glass substrates (anyway, the authors should try). (3) CB was developed for periodic structures – such periodicity is not visible from Fig 2. The surface with more or less random nodular structures rather suggests a type of roughness.
It seems that an effect other than a CB situation is responsible for the contact angles measured (when in fact increased).
Results and discussion, optical measurements, page 7 and 8
- End of second par, Fig. 4b, light blue line: why should the valley be caused by reflection of light since the width of the ‘lines’ is in the order of the incident wavelength? It is unlikely, that the 30%/120 min sample provides this situation, but the 90 min and 150 min do not. Evaluation of linewidth would be required to support this argument.
- Fig 4 a: according to this measurement, the 30%/180 min really seems to be special. It is ultimately needed to add at least two AFM (or SEM) micrographs, namely for the 30%/90 min and 30%/180 min sample, those with the smallest and highest contact angle
- Page 7, bottom: this is equ 3!
- Fig 5: Potentially it is advised to present the bars with a suppressed zero, e.g. only from 70-100%, in order to highlight the differences (if desired). When the message is rather “there is no large difference” then the actual scaling makes sense. However, the text should address one of the two perceptions (differences or similarity). AND: skip ‘missing’ bar in Fig 5a – a bar chart does not require a linear axis.
- Caption of Fig 5, missing conditions: 180 min for a, 30% for b.
- Comparison to Baquedano: transmittance increases with lines, but decreases with random structures – the structures shown here are rather random than linear, with heights between 20 and 200 nm! There is no clear argument that the height is responsible for the difference rather than the ‘order’.
References
- Numbering is duplicate
- Correct title of CB is “Wettability of porous surfaces”
The following is suggested to the authors wrt a prospective procedure to improve the manuscript.
- First, collect AFM (or SEM) measurements to characterize all samples in a reproducible way. In particular, micrographs for 30%/180 min and 30% 90 min should be added to the manuscript.
- Evaluate all roughness values from AFM.
- Confirm contact angles (see issues below) by measurements at differing locations or with different samples. Consider potential further influencing issues (side view: directionality? (lines), top view: circularity? Uniformity of contour? (contamination/particles)
- Potentially the authors have to look for literature other than Cassie-Baxter to explain their results (in case that the increase of contact angle is confirmed) – see list of references below.
- In view of potential ‘other’ reasons identified the structures should become re-examined and possibly re-characterized (undercuts? hierarchical structures?, …)
- Then, with a clear view on the structures AND the contact angles, re-formulate the theoretical part (see issues above).
- If necessary/advised, repeat optical characterization under potential novel aspects that have incurred.
To support the authors the following papers may be helpful:
- Cao et al, “Design and fabrication of micro-textures for inducing a superhydrophobic behaviour on hydrophilic materials”
- Nosonovsky et al: “Hierarchical roughness makes superhydrophobic states stable”
- Herminghaus: “Roughness-induced non-wetting”
- Marmur: “From hygrophilic to superhygrophobic: theoretical conditions …”
- Tuteja: “Robust omniphobic surfaces” and “Designing superoleophobic surfaces”
- Dufour et al: “Engineering sticky superomniphobic surfaces …”
- Chou et al: “Wetting behaviour of a drop atop holes
These references are NOT the clue to explain your experiments, but nonetheless may be helpful to understand the issues addressed above.
In case your results are correct, the problem is that even with an increase of the contact angle the surface is not in a hydrophobic state (and by far not a superhydrophobic one) – thus an adequate reference is hard to find.
Hopefully the authors will enjoy when reading.
Contact angles and wetting configurations are a stimulating field, though not easily attained
(and also not yet completely understood).
Reviewer 2 Report
Comments and Suggestions for Authors
This study presents a method for nanostructuring soda-lime glass by combining soft lithography with KOH wet etching, allowing for control over the glass's wetting properties. By adjusting the KOH concentration and etching time, contact angles ranging from superhydrophilic (~9°) to hydrophobic (~60°) were achieved, similar to results obtained through reactive ion etching. The process minimally affects transparency, maintaining optical transmittance above 90%, making it suitable for applications like photovoltaics, where surface texturing and transparency are crucial. In my opinion, acceptance of this manuscript is recommended after a satisfactory response to the following points:
- As a reader, more information on what "soft lithography" refers to in the manuscript would be helpful.
- It is recommended that the curves in Figure 4 be smoothed to facilitate the reader's evaluation. Additionally, the description of the test equipment model and light source needs further clarification.
- The environmental temperature is quite important for the wet etching process. Therefore, more careful and complete information is needed in the description of Figure 5.
- It is recommended that the potential applications of this methodology be described in the conclusions section.
Minor editing of English language required
Round 2
Reviewer 1 Report
Comments and Suggestions for Authors
Comments on revised manuscript nanomaterials 3156794-v2
The authors have addressed some of the critical issues addressed in the comments, and have improved parts of the manuscript.
However, there are still severe weak points remaining to be revised before publication of the manuscript.
The authors are explicitly asked and invited to implement the additional comments below in their text.
In particular the comment on response 16 seems to be suited to improve the reasoning of the manuscript substantially.
The methods section is much better now.
Still missing in the text: approximate thickness of PDMS stamp as well as its size - see Comment 2 F, Response 3E.
Further detailed issues:
End of part 3.1
The authors should implement their estimation, that the width variation observed after KOH etching is most probably due to a limited mask selectivity, as the PMMA is partly attacked by KOH (and the PMMA, determined by soft lithography, features some ‘native’ height (and width?) modulation). – see response 4D.
(By the way, KOH is a base, not an acid – see response 4C)
Response 5 + 14
The simulation model used has to be implemented in the text, similarly the averaging and the 3 different places!
(could go to the methods section, page 3)
Response 8
I understand the issue with the glass SEM.
But why did the authors not use an AFM image?? (imageJ could identify a period across the lines, even when the lines do resemble more to mounds than to lines)
AND: now, with the proper experimental conditions at hand (in the caption: 20%, 180 min) the question arises, why the authors did not take a DIFFERENT sample to characterize the lines.
It is most probable, that the ‘strange’ result is obtained due to the long etching time, and that choice of a shorter one would be more credible/convincing to clearly identify lines.
The ‘lines’ (or rather bumps) of Fig. 2 are still unexplainable, which weakens the publication substantially.
It would be much better to choose another sample, especially as the present sample of Fig. 2 does NOT correlate with any specific situation within the rest of the results.
Response 9
OK.
However, it was suggested to add an additional scan in Fig. 1c/d, in order to show the initial situation before etching (how the MASK height varies locally!)
Response 15
I understand that time is money.
Nevertheless, confirming the measurements does NOT explicitly mean to prepare additional samples. It is e.g. an option to repeat the contact angle measurements taking the samples existing – this should be possible with a quite limited effort, and could really corroborate the data basis.
At the premises given, this should be done by all means.
P 6 par 2, (12 lines) and Comment/response 17
The authors have exchanged their wrong reference with another one of the same researcher – sorry, it was not intended to mis-acknowledge a female researcher.
However, the new reference is still not cited correctly.
Baquedano did NOT vary the etching time, it stayed at 1 min.
The aspect ratio showed some correlation with the contact angle (as mentioned here also), however the aspect ratio did NOT increase with increasing RIE power.
And, still, it is NOT possible to compare an RIE etching process (combining an ionic attack with a chemical attack) with a KOH etching, which is purely chemical. (though some similarities with the results, the authors have to address this difference and discuss it critically)
Furthermore, Baquedano used an SiO2 hardmask, NOT a PMMA mask (they investigated just the second step, after patterning the hard mask by means of the PMMA).
And finally, KOH concentration may be regarded as some correspondence with the RIE power (as it changes the etch rate), however, etching time does NOT increase the etching ‘intensity’ - whatever this should be; ‘intensity’ is NOT a term suitable to characterize an etching process. Etching time is vital for the structures obtained, as a limited mask selectivity may change the etching situation.
AND, most important:
On page 8 (bottom, “Wettability”), after addressing the different situation with Wenzel and Cassie-Baxter based on literature, Baquedano clearly states, that “for all surfaces, the wetting state corresponds to a Wenzel state and the contact angle never exceeds the critical angle needed for the transition to a Cassie-Baxter condition”.
This is exactly the content/meaning of the respective paragraph in the initial comments on Wenzel and Cassie-Baxter!!!!! (Comments 19/20)
Cassie-Baxter is NOT APPLICABLE for the results reported in the current manuscript!
The respective paragraph on page 7 (though already somewhat alleviated in the present text) CANNOT remain as it is. The critical angle (the minimum required) to achieve a CB state is above 90 deg, which is not the case here! (Please see the references noted in the comments on V1)
The authors may e.g. state, that the results (though the contact angle increased) cannot be explained by CB, as the highest contact angle of 61 deg is still too small to exceed a critical value required for CB (which is > 90 deg)
Response 16
This is interesting!
When you measure a roughness by AFM (a DIRECT method), and when this roughness (when I understood the authors correctly, in terms of an area ratio it amounts to r = 1.003) is far from any roughness value (r = 1.1) that can be evaluated from the contact angle (which is an INDIRECT method, related to the Wenzel theory), then this Wenzel theory is NOT APPLICABLE to the samples. The authors should contrast these differing roughness values in the text!
This brings new light into the strange experimental situation!
CB is NOT applicable, due to the size of the contact angle (too low, critical value not exceeded), but neither can the results be explained by Wenzel (as roughness is far from corresponding to AFM measurements)
THIS IS A RESULT – though unexpected!
In addition, the holes clearly visible in the AFM in the authors response also demonstrate, that the samples do NOT feature a roughness of the Wenzel type.
AND: the presence of holes MIGHT INCREASE the contact angle! (see e.g. the additional references of the comments on V1)
In order to bring the manuscript into a form that is scientifically credible, it is proposed to bring these two AFM figures of the authors’ response into the paper, and address them in context with the detected wettability change (the authors may address Wenzel and CB (as they already have), but then critically discuss that both theories are NOT SUITED to explain their results (based on values for AFM roughness measured, and, also on r < 1 values, see response 19), and potentially address the holes to be a possible reason for the results obtained.
Response 22
In view of the above paragraph, it seems that the implementation of the SEM micrographs of the new Fig. 5 is less needful, particularly as it does not help to a deeper understanding.
Moreover, the second micrograph (once again) renders recognition of a line pattern at the surface barely possible.
Response 27
The authors state, that “the aim of this manuscript is to show the possibility of glass surface nanostructuring in a simple way” – why did they then bring physics into play with Wenzel and Cassie-Baxter, however, in a way that is NOT scientifically correct??
I know, that publication of research is important, to motivate researchers and to justify funding. But in doing so, science should not ‘fall by the wayside’.
Without convincing/demonstrative results it becomes hard (as also the alchemists had to learn).
In my opinion, beyond the responsibility towards (young) scientists and funding a scientist also has a responsibility towards a correct and credible physics (implying some depth of contention of the respective theory before using it, potentially disregardful) – this is just my opinion, it does not imply a criticism of the authors’ praxis (I am not aware of the specific situation prevailing and of the compromises required)
All things considered, reasoning along response 16 (above) seems to be well-suited to bring this manuscript into a shape that is suitable for publication.
Round 3
Reviewer 1 Report
Comments and Suggestions for Authors
Comments on second revision of manuscript nanomaterials 3156794-v3
Though not all modifications of the second version are fully convincing, the manuscript has now achieved a state that is near to appropriate for publication.
In particular, the main text with the revised section on CB and W is now of scientific relevance, thus overcoming the key aspects prohibiting a potential disclosure.
However, the authors have lost site for revision of two further passages, which were not addressed in the comments explicitly, though rather self-evident: The modification required in the ABSTRACT and in the CONCLUSION according to the novel/amended scientific conception with respect to CB and W in the main text. THIS ADAPTION IS ULTIMATELY REQUIRED !
In addition, the transition from Baquedano’s results to W and CB on page 6. It seems, that an additional sub-title there could eliminate that issue, e.g.:
3.2 Wetting properties
3.2.1 Experimental results
3.2.2 Theoretical aspects
